# Roles of microRNAs in Hepatitis C Virus Replication and Pathogenesis

**DOI:** 10.3390/v14081776

**Published:** 2022-08-15

**Authors:** Hui-Chun Li, Chee-Hing Yang, Shih-Yen Lo

**Affiliations:** 1Department of Biochemistry, Tzu Chi University, Hualien 97004, Taiwan; 2Department of Laboratory Medicine and Biotechnology, Tzu Chi University, Hualien 97004, Taiwan; 3Department of Laboratory Medicine, Buddhist Tzu Chi General Hospital, Hualien 97004, Taiwan

**Keywords:** hepatitis C virus, microRNAs, replication, pathogenesis, biomarker

## Abstract

Hepatitis C virus (HCV) infection is associated with the development of chronic liver diseases, e.g., fibrosis, cirrhosis, even hepatocellular carcinoma, and/or extra-hepatic diseases such as diabetes. As an obligatory intracellular pathogen, HCV absolutely relies on host cells to propagate and is able to modulate host cellular factors in favor of its replication. Indeed, lots of cellular factors, including microRNAs (miRNAs), have been identified to be dysregulated during HCV infection. MiRNAs are small noncoding RNAs that regulate protein synthesis of their targeting mRNAs at the post-transcriptional level, usually by suppressing their target gene expression. The miRNAs dysregulated during HCV infection could directly or indirectly modulate HCV replication and/or induce liver diseases. Regulatory mechanisms of various miRNAs in HCV replication and pathogenesis have been characterized. Some dysregulated miRNAs have been considered as the biomarkers for the detection of HCV infection and/or HCV-related diseases. In this review, we intend to briefly summarize the identified miRNAs functioning at HCV replication and pathogenesis, focusing on the recent developments.

## 1. Introduction

Liver cancer was the sixth most common cancer and also the third leading cause of cancer death worldwide in 2020 (https://gco.iarc.fr/today/home, accessed on 24 May 2022). Hepatocellular carcinoma (HCC) represents around 90% of all cases of primary liver cancer. Hepatitis C virus (HCV) infection is a risk factor associated with the HCC development. HCV, a hepatotropic virus, belongs to the hepacivirus genus of the Flaviviridae family. HCV infection could cause acute or chronic hepatitis. Actually, 50–80% of HCV patients develop asymptomatic chronic infection. According to the World Health Organization (WHO), approximately 71 million individuals worldwide are chronically infected by the HCV (https://www.who.int/hepatitis/publications/global-hepatitis-report2017/en/, accessed on 24 May 2022). If left untreated, patients with chronic hepatitis C (CHC) are at high risk of developing progressive fibrosis, liver cirrhosis and even HCC. Previous studies also showed that HCV plays a positive role in the development of extrahepatic diseases, e.g., diabetes. Although the majority of CHC can now be cured using various direct-acting antivirals (DAA), a successful treatment does not guarantee no further HCV reinfection. In addition, the high mutation rate of HCV may lead to the development of DAA resistance. Thus, HCV eradication will probably require an effective vaccine. HCV is classified into seven major genotypes (1 to 7) because HCV genomic RNA sequences are highly heterogeneous among different isolates [1]. The heterogeneity of HCV genome hinders an effective vaccine development against infection from all HCV genotypes (for a review, see [2]).

The life cycle of HCV begins with a virion binding to its specific entry factors (or receptors) on hepatocytes. Then, the virion enters the cytoplasm via endocytosis and releases its genomic RNA. The HCV single-stranded, positive-sense RNA genome is about 9.6 kb containing a conserved 5′ untranslated regions (UTR), one open reading frame (ORF) and a conserved 3′-UTR. The HCV genomic RNA is used for viral polyprotein translation and also viral replication. The 5′UTR contains the internal ribosome entry site (IRES) that recruits ribosomes to initiate the translation. The single HCV ORF encodes for a polyprotein of approximately 3010 amino acids, which is cleaved by cellular and viral proteases into structural (core, E1 and E2) and nonstructural (p7, NS2, NS3, NS4A, NS4B, NS5A and NS5B) proteins. The sequence elements and secondary structures in the 5′UTR and core coding region are also involved in the replication of HCV. The 3′UTR is a specific tripartite structure—a variable region, a poly (U/UC) tract, and a highly conserved X tail—required for efficient viral RNA replication. The genomic RNA replication of HCV occurs within the membranous web in the endoplasmic reticulum (ER). HCV genomic RNAs are packaged by viral core proteins and enveloped by a lipid membrane with two viral glycoproteins (E1 and E2) to form the virions. Finally, various viral proteins and cellular proteins (e.g., the very-low-density lipoprotein (VLDL) biosynthetic pathway) are required to assemble and release the HCV virions from cells (for a review see [3]).

Approximately 70–90% of the human genome is known to be actively transcribed. However, only around 2% of these transcripts are translated into proteins while more than 90% of them are noncoding RNAs (ncRNA). These ncRNAs can be classified into small ncRNAs (shorter than 200 nucleotides (nt)) and long ncRNAs (lncRNAs, longer than 200 nt) based on size. Micro-RNAs (miRNAs) are one kind of small ncRNAs. Several processing steps are required to generate the mature miRNAs. The initial transcripts of miRNAs genes catalyzed by RNA polymerase II are the primary miRNA transcripts (pri-miRNAs). Pri-miRNAs containing the 5′cap and 3′ poly-A tail are then cleaved by Drosha and DCGR8 to form precursor miRNAs (pre-miRNAs, 70 to 100 bp) in the nucleus. Then, pre-miRNAs are transported to the cytoplasm via Exportin 5 and further cleaved by Dicer to produce mature miRNA duplexes with 3′-overhangs (~22-bp). After strand separation, only the single-stranded mature miRNA is recruited into the RNA-induced silencing complex containing Ago2 proteins and recognized by its target mRNAs. Typically, miRNAs regulate eukaryotic gene expression at the post-transcriptional level through specific base-paring interactions between the miRNA’s 5′ end (“seed” region) of and the target mRNAs’ coding or 3′UTR regions. MiRNA exerts its function by inhibiting the target mRNA translation (when only partial pairing between miRNA and its target mRNA) or degrading the target mRNAs (when there is a perfect base pairing between miRNA and its target mRNA) (for a review see [4]).

The human genome expresses over 2600 miRNAs [5] (www.mirbase.org/, accessed on 24 May 2022). Each miRNA can regulate numerous target mRNAs and every mRNA is likely to be regulated by several different miRNAs. Thus, it is believed that miRNAs are involved in almost all biological process such as development, immune response, aging, cell proliferation and apoptosis. Accumulated evidences also demonstrate that cellular miRNAs can interact with viruses at multiple levels to modulate viral replication [6].

HCV depends on its host cells to infect successfully. High-throughput sequencing data did not identify HCV-encoded miRNAs [7], suggesting that HCV genome is unlikely to encode any miRNAs. On the other hand, accumulated evidence showed that HCV infection modulates the expression of many cellular miRNAs, which in turn regulate HCV replication directly or indirectly [8]. Moreover, hepatocytes have been shown to guard against HCV infection by modifying some miRNAs expression. In this review, we intend to summarize the miRNAs modulating HCV replication either through direct interaction with viral genomic RNA or indirect modulation of cellular factors important for viral replication, such as interferon (IFN) signaling pathway. In addition, we briefly describe the roles of miRNAs in HCV-related liver disease progression.

## 2. Roles of miRNAs in HCV Replication

HCV relies on the cellular components in host cells to survive and propagate. Many studies have identified miRNAs differentially expressed during HCV infection [8]. These dysregulated miRNAs have diverse roles in modulating HCV infection [9]. Some cellular miRNAs suppress while others facilitate viral infection [10]. The life cycle of HCV can be divided into stages of viral entry, protein translation, genome replication and viral assembly/release [3]. We will describe the miRNAs involved in each stage in sequence (Figure 1) (Table 1). Each HCV genotype has different disease outcomes and also a different sensitivity to the therapeutic treatment, e.g., interferon and/or even DAAs [1]. Various cellular miRNAs should have different effects on the life cycle of different HCV genotypes, e.g., expression of miR-122/155 was different in each genotype [11]. However, in most studies, effects of miRNAs on different HCV genotypes were not specifically characterized. Thus, roles of miRNAs in HCV replication did not focus on certain HCV genotypes.

### 2.1. miRNAs Modulating HCV Entry

To infect a hepatocyte, HCV virions first have to interact with several cell surface receptors and co-receptors, e.g., heparan sulfate proteoglycans (HSPGs), cluster of differentiation 81 (CD81), claudin-1 (CLDN1) and occludin (OCLN) [3]. MiRNAs affecting the expression of these receptors and/or co-receptors would modulate HCV entry.

#### 2.1.1. miR-548m and miR-194

Epigallocatechin gallate (EGCG) is a powerful antioxidant with antiviral effects. EGCG was found to upregulate miR-548m expression in cultured cells. Two miR-548m binding sites were identified in the CD81 mRNA 3′UTR. Indeed, miR-548m could repress CD81 expression and thus reduce the HCV infectivity. These results suggest that EGCG could act as an anti-HCV agent through increasing miR-548m expression and suppressing CD81 receptor expression [12]. The miR-194 was also found to hinder HCV entry through targeting CD81 receptor [13].

#### 2.1.2. miR-182

CLDN1 mRNA 3′UTR has binding sites for miR-182. As expected, miR-182 could inhibit CLDN1 expression and reduce the HCV viral load [14]. 

#### 2.1.3. miR-122 and miR-200c

miR-122 and miR-200c target the 3’-UTR of OCLN mRNA and reduces the OCLN expression. Overexpression of miR-122 and miR-200c can suppress HCV entry into hepatocytes [15,16]. 

### 2.2. miRNAs Modulating HCV Translation and/or Genome Replication

Once in the cells, the HCV genome will be released into the cytosol. The single-stranded, positive sense HCV genomic RNA can be the template for viral replication and/or as mRNA for the viral protein expression. Therefore, cellular miRNAs can modulate the HCV replication and/or translation via direct binding with HCV genomic RNA (Figure 2). Alternatively, miRNAs can affect the virus replication through regulating the cellular factors participated.

Several miRNAs have been found to interact with HCV genome directly and inhibit HCV replication (Figure 2). 

#### 2.2.1. miR-196, miR-296, miR-351, miR-431 and miR-448

Several IFNβ-induced miRNAs (miR-196, miR-296, miR-351, miR-431 and miR-448) had almost perfect complementarity in their seed sequences with HCV genomic RNA. Transfection of these miRNAs individually revealed that they were indeed able to substantially suppress HCV replication [17]. miR-448 targeted the core coding region while miR-196 interacted NS5A coding region of the HCV RNA genome [17]. 

#### 2.2.2. miR-199a

miR-199a has been reported to inhibit HCV replication through directly targeting the stem-loop II region of 5′UTR of HCV genome when overexpressed, while its silencing enhanced HCV viral replication [18]. However, another report has demonstrated that suppression of miR-199a-5p reduced HCV replication via regulating the pro-survival pathway [24]. Further studies are required to clarify this issue.

#### 2.2.3. Let-7b

Let-7b inhibits HCV replication through directly targeting the NS5B coding region and the 5′ UTR sequences of the HCV RNA genome, which are conserved among genotypes [19]. 

#### 2.2.4. miR-181c

miR-181c targeted E1 and NS5A regions of HCV genome and its overexpression suppressed viral replication. On the other hand, downregulation of miR-181c was detected by HCV infection or NS5A protein via regulation of CCAAT/enhancer binding protein β (C/EBPβ) [20].

#### 2.2.5. miR-122

The most abundant miRNA in liver is the liver-specific miR-122 [21], which is around 60–70% of the total miRNA in hepatocytes. MiR-122 has been demonstrated to play a critical role in liver homeostasis and hepatocarcinogenesis [25]. Many studies have reported that HCV—and also equine hepacivirus—replication requires miR-122 in infected cells [26,27,28]. Moreover, miR-122 expressed exogenously allows more efficient virus replication in nonpermissive cell line [29]. The role of miR-122 in HCV replication could be compensated by various miRNAs [30]. Direct annealing of miR-122 with two adjacent sites in the 5′UTR of HCV genomic RNA [31] has been shown to (a) stimulate the translation of viral protein synthesis by promoting the formation of IRES RNA structure [32,33,34], (b) stabilize the genome to mask and protect the 5′end from the degradation mediated by DUSP11 (or DOM3Z) andXRNs [35,36,37]; and (c) induce viral genomic RNA replication [26,38,39]. Seven more miR-122 binding sites on the HCV genome were predicted using bioinformatics method: four in the NS5B coding region and three in the 3′UTR. However, mutational studies have demonstrated that these seven potential miR-122 binding sites did not impact HCV promotion [40]. Thus, the known functional binding sites for miR-122 were the two sites in 5′UTR (Figure 2). Through interacting with miR-122, HCV and equine hepacvirus RNAs could sequester miR-122 and miR-122-targeted cellular mRNAs were de-repressed during virus infection [28,41].

### 2.3. miRNAs Modulating HCV Assembly

An HCV genomic RNA and the viral structural proteins are the required components for virus particles assembly. Other HCV nonstructural proteins and host cellular factors are also essential for particles assembly, for example, the factors involved in the VLDL synthesis and secretion. Cellular miRNAs could modulate HCV assembly through the regulation of these cellular factors involved in viral assembly.

#### 2.3.1. miR-99a

Downregulation of miR-99a expression by HCV infection is reported [22]. Transfection of miR-99a-5p mimics suppressed the expression of mammalian target of rapamycin (mTOR) and sterol regulatory element binding protein (SREBP)-1c in Huh-7 cells, and it also led to reduced lipid accumulation in cells after oleic acid-treatment, which resulted in a decreased level of HCV RNA. Thus, miR-99a reduces intracellular lipid accumulation by modulating mTOR/SREBP-1c and leads to suppression of HCV replication and packaging [22]. 

#### 2.3.2. miR-501-3p and miR-619-3p

O-linked N-acetylglucosamine (O-GlcNAc) transferase (OGT), the target of miR-501-3p and miR-619-3p, is the regulatory cellular factor for HCV assembly. Both miRNAs were found to downregulate OGT protein expression. OGT-silencing resulted in a significant enhancement in the HCV infectivity [23].

## 3. MiRNAs Involved in the Interferon Pathways

Interferon (IFN) activates Janus kinase/signal transducer and activator of transcriptions (JAK/STAT) pathway leading to the expression of IFN-stimulated genes (ISGs) and elicits an anti-viral response. STAT1, STAT2 and interferon regulatory factor 9 (IRF9) could form a complex to trans-activate ISGs. On the other hand, JAK/STAT signaling is inhibited by suppressors of cytokine signaling (SOCS) proteins such as SOCS3 [42]. Therefore, both JAK/STAT pathway and SOCS proteins are important regulators in IFN response of the HCV-infected patients. Furthermore, HCV is able to evade the IFN response by modulating various cellular miRNAs [43] (Table 2).

### 3.1. miRNAs Inhibiting IFN Production and/or Signaling 

#### 3.1.1. miR16

Compared with healthy controls, miR16 expression was found to be significantly upregulated while expression of IRF3 and small mother against decapentaplegic homolog 7 (SMAD7) was significantly downregulated in HCV patients. Therefore, the interaction between IFN and TGFβ pathways via IRF3 and SMAD7 in the immune response against HCV infection could be regulated by miR16 [44]. 

#### 3.1.2. miR-21

The miR-21 expression is increased by HCV infection. Both HCV NS5A and NS3/A4 proteins can enhance activator protein 1 (AP-1) binding to the miR-21 promoter, which upregulate miR-21 expression. Then, miR-21 suppressed the type I IFN expression and the subsequent anti-viral response. In addition to inhibiting NF-kB, mR-21 was shown to target myeloid differentiation factor 88 (MyD88) and interleukin-1 receptor-associated kinase 1 (IRAK1), which are essential for the activation of IRF7, a master regulator of IFNα signaling [45]. 

#### 3.1.3. miR-208b and miR-499a-5p

Expression of two miRNAs, miR-208b and miR-499a-5p, was induced by HCV in hepatocytes. These two miRNAs support viral persistence by suppressing IFNL2 and IFNL3, which are members of the type III IFN gene family. Moreover, miR-208b and miR-499a-5p also attenuate type I IFN signaling by reducing the IFN receptor 1 (IFNAR1) expression directly in HCV-infected hepatocytes [46]. 

#### 3.1.4. miR-93-5p

It has been shown that HCV core protein upregulated miR-93-5p expression, which in turn inhibited the IFN signaling pathway through targeting IFNAR1 directly. The miR-93-5p induced attenuation of IFN signaling allows HCV replication in infected cells [47]. 

#### 3.1.5. miR-373

HCV infection stimulates miR-373 expression [48]. miR-373 was found to impair JAK/STAT signaling and reduce IFN-induced antiviral responses by targeting JAK1, IRF9 and also IRF5 [48,49]. 

#### 3.1.6. miR-135a

miR-135a expression is upregulated by HCV infection. miR-135a was reported to preferentially enhance HCV replication. Bioinformatics analyses and functional assays have shown that three antiviral host factors, including receptor interacting serine/threonine kinase 2 (RIPK2), MYD88, and C-X-C motif chemokine ligand 12 (CXCL12) are the targets of miR-135a [50]. 

#### 3.1.7. miR-758

miR-758, involved in lipid and cholesterol metabolism, is highly expressed in the liver. Expression of miR-758 was upregulated by HCV infection or HCV core protein. It has been shown that overexpressed miR-758 targeted and then suppressed Toll-like receptor 3 (TLR3) and TLR7, leading to the reduction in type I IFNs and dampened IFN signaling [51]. 

#### 3.1.8. miR-125a

Expression of miR-125a was upregulated by HCV infection in hepatocytes. miR-125a was found to target and suppress the expression of mitochondrial antiviral signaling (MAVS) and TRAF6, important for the IFN signaling pathway. As expected, disrupting the expression of these two genes can compromise type I IFN responses to HCV infection. Thus, HCV can evade immune surveillance by activating miR-125a [52]. 

#### 3.1.9. miR-942

HCV infection led to cell apoptosis mediated by ISG12a, one of ISGs. Induction of Noxa contributed to ISG12a-mediated apoptosis during HCV infection. ISG12a was predicted as the target for miR-942 by bioinformatics search. Further studies showed that HCV suppressed miR-942 expression and the miR-942 level was inversely correlated with the ISG12a expression. Moreover, exogenous expression of miR-942 reduced ISG12a expression in cells and subsequent inhibition of apoptosis triggered by HCV infection. Thus, miR-942 modulates HCV-induced apoptosis of hepatocytes through targeting ISG12a [53].

### 3.2. miRNAs Enhancing IFN Production and/or Signaling

#### 3.2.1. miR-221

miR-221 inhibited the expression of SOCS1 and SOCS3, which suppress IFN-α activity. Accordingly, overexpression of miR-221 could increase IFN-mediated anti-HCV response [54]. 

#### 3.2.2. miR-30

Treatment with IFN-α/β in hepatocytes regulates the expression of HCV-specific miRNAs including the miR-30 cluster, miR-342-5p, miR-489, miR-142-3p, and miR-128a. The miR-30 cluster targets SOCS1 and SOCS3, negative regulators of cytokine signaling, and it modulates the differentiation of helper T cell and also the JAK/STAT signaling pathway [55].

#### 3.2.3. Let-7b

Let-7b targeted and reduced the SOCS1 protein and then increased the expression of IFN downstream ISGs. Let-7b also targeted the ATG12 and IκB kinase alpha (IKKα) mRNAs and inhibited the interaction of the ATG5-ATG12 conjugate and retinoic acid-inducible gene I (RIG-I) resulting in enhanced IFN expression, which could further facilitate JAK/STAT signaling [56]. 

#### 3.2.4. miR-122

The activation of IFNs in response to HCV infection is significantly enhanced by increasing miR-122 levels in hepatoma cells. Mechanistically, miR-122 targeted three receptor tyrosine kinases, i.e., MERTK, FGFR1 and IGF1R, which enhance STAT3 phosphorylation directly. Thus, miR-122 suppressed the phosphorylation of STAT3, thereby reducing the negative effect of STAT3 on IFN-signaling [57]. 

#### 3.2.5. miR-29c

Expression of miR-29c was downregulated in HCV infected cells. MiR-29c targeted and suppressed STAT3 expression. HCV replication was repressed by overexpression of miR-29c or knockdown of STAT3, while promoted by depletion of miR-29c or upregulation of STAT3. Furthermore, the miR-29c-mediated suppression of HCV replication can be reversed by STAT3 overexpression. Moreover, the anti-miR-29c-mediated suppressive effect on type I IFN response was reduced after STAT3 knockdown. Thus, miR-29c might suppress HCV infection through enhancing type I IFN response via targeting STAT3 [58].

### 3.3. miRNAs Induced by IFN Reduce HCV Replication

Several IFN-induced miRNAs, such as miR-1, -30, -128, -196, -296, -351, -431, -448, -324-5p and miR-489, have been shown to suppress viral replication [59,60,61]. On the other hand, HCV infection represses the expression of some of these miRNAs. 

## 4. miRNAs Modulating Other Cellular Factors Involved in HCV Replication

In addition to IFN signaling, other pathways and/or factors are also targeted by miRNAs dys-regulated by HCV infection (Table 3).

### 4.1. miRNAs Suppressing HCV Replication

#### 4.1.1. miR-125b-5p

HuR could enhance HCV replication. miR-125b-5p reduces HCV replication by targeting HuR. Reduction of miR-125b-5p upregulated HuR protein expression and then enhanced HCV replication [62].

#### 4.1.2. miR-196a

In addition to directly binding to the HCV RNA genome, miR-196a targets the 3′-UTR of Bach1 mRNA and reduces Bach1 expression, a transcriptional repressor of heme oxygenase-1 (HO-1). The de-repression of HO-1 by miR-196a inhibits HCV replication. HO-1 possesses anti-inflammatory and anti-oxidant activities [60].

#### 4.1.3. Let-7c

Overexpressing miR-let-7c could induce HO-1 expression through targeting its transcriptional repressor Bach1 and markedly reduce HCV replication. In contrast, the antiviral activities of miR-let-7c were abated by miR-let-7c inhibitor treatment, overexpressing Bach1 or inhibiting HO-1 activity [63].

#### 4.1.4. miR-503

Cytokine tumor necrosis factor-α (TNF-α) mediates the immune response to viral infections, such as HCV. HCV NS5A overexpression suppressed TNF-α-induced cellular apoptosis through modulating miR-503 expression. Nuclear factor kappa-light-chain-enhancer of activated B cells (NF-κB) could bind the miR-503 promoter and promote the miR-503 transcription. MiR-503 targeted and reduced B-cell lymphoma 2 (BCL-2) expression. Thus, NS5A protein suppresses miR-503 expression by inhibiting NF-kB activation, and thus, it enhances BCL-2 expression to reduce apoptosis in hepatocytes [64].

#### 4.1.5. miR-181c

Expression of miR-181c was downregulated by HCV infection through C/EBP-β in hepatocytes [17]. A direct binding of miR-181c to the 3′ UTR of ataxia-telangiectasia mutated (ATM) mRNA has been demonstrated. ATM, a protein kinase, is a central mediator for the response to cellular DNA double-strand break. Exogenously overexpressing miR-181c reduced ATM expression and promoted apoptosis of HCV infected hepatocytes. Thus, miR-181c could suppress HCV replication [65].

#### 4.1.6. miR-130a

Contradictory results about the interactions between miR-130a and the host innate immune system were obtained [66,67]. In addition, it has been reported that HCV replication is downregulated by miR-130a via two independent pathways: enhancement of host immune responses and modulation of pyruvate metabolism. ATG5 is one of the target genes for miR-130a. ATG5 would downregulate the expression of ISGs and upregulate HCV replication significantly. MiR-130a modulates host antiviral response and HCV replication by targeting ATG5. Moreover, miR-130a regulates HCV replication and subsequent pyruvate production via targeting pyruvate kinase in liver and red blood cell (PKLR) [68,69].

#### 4.1.7. miR-27a

HCV replication occurs in the membranous web derived from ER. Formation of the membranous web is dependent on lipid metabolism. Thus, modulation of lipid metabolism is critical for efficient HCV replication. HCV induces the expression of miR-27a [70,71], which in turn reduces the expression of many genes involved in lipid metabolism including ApoA1, ApoB100 and ApoE3, thus reducing the production of infectious HCV particles [72]. Therefore, miR-27 is involved in a self-inhibitory mechanism to ensure an appropriate HCV level for a persistent infection. Moreover, miR-27 downregulates angiopoietin-like protein 3 (ANGPTL3) [71] to increase lipoprotein lipase activity, which has been identified as an anti- HCV cellular factor involved in the inhibition of HCV replication [73].

#### 4.1.8. miR-185

The effect of miR-185-5p on HCV replication has been studied recently. Downregulation of miR-185-5p was detected in HCV-infected cells. Overexpressed miR-185-5p suppressed HCV replication. GALNT8 is the target of miR-185-5p. Moreover, the effects of upregulation or downregulation of miR-185-5p on HCV replication were correspondingly abated by the overexpression or knockdown of GALNT8. Thus, miR-185-5p may target GALNT8 and then inhibit HCV replication [74].

#### 4.1.9. miR-29

MiR-29 overexpression reduced HCV replication [75]. The mechanism is not yet known.

### 4.2. miRNAs Facilitating HCV Replication

#### 4.2.1. miR-141

miR-141, upregulated by HCV infection, is required for efficient HCV replication. Indeed, overexpression of miR-141 increased HCV replication, while depletion of miR-141suppressed HCV replication. One of the target genes of miR-141 is deleted in liver cancer 1 (DLC-1). Increasing the amount of miR-141 decreased DLC-1 protein but not mRNA level, suggesting that miR-141 is involved in translational suppression of DLC-1 [76]. 

#### 4.2.2. miR-320c, miR-483

Expression of miR-134, miR-320c and miR-483-5p was found to be significantly upregulated in the serum of HCV infected patients. Indeed, HCV infection enhances the expression of miR-320c and miR-483-5p. The phosphatidylinositol 3-kinase (PI3K/Akt), mitogen-activated protein kinase (MAPK) and NF-κB signaling pathway were predicted to be targeted by miR-320c and miR-483-5p. Thus, these two miRNAs may play roles in immune evasion and cell survival [77]. 

#### 4.2.3. miR-122

MiR-122 binds directly to HCV RNA genome, and it also indirectly facilitate HCV replication via Bach1-mediated downregulation of HO-1 [78]. 

#### 4.2.4. miR-491

miR-491, which is downregulated by HCV infection, has been known to help HCV entry and thus augment viral replication in HCV-infected cells. This miR-491 induced replication augmentation of HCV is dependent on the PI3K/Akt pathway because it is abolished in the presence of PI3K inhibitor [79].

## 5. Roles of miRNAs in HCV-Related Diseases

HCV infection often induces chronic liver inflammation and causes hepatocyte injury such as liver fibrosis, cirrhosis and even HCC. These HCV-related diseases are the indirect consequence of host immune responses to HCV infection and/or directly caused by viral proteins induced deregulation of cellular metabolism. The detailed mechanisms of how HCV infection leads to HCV-related liver diseases remain elusive. However, the interactions between cellular factors, including miRNAs and viral components, have been demonstrated in the development of these HCV-related liver diseases [6]. 

### 5.1. miRNAs Modulating Inflammation

#### 5.1.1. miR-449a and miR-107

Two cellular miRNAs, miR-449 and miR-107, which are downregulated by HCV infection, result in increased inflammatory responses. It has been shown that miR-449a targets NOTCH signaling pathway after HCV infection and regulates the level of YKL40, an inflammatory marker [80]. 

#### 5.1.2. miR-155

In HCV treatment-naïve patients, increasing miR-155 amount was detected in circulating monocytes, which caused the augment of TNF-α production [81]. In vitro studies with human monocytes showed that expressed HCV proteins such as core, NS3, and NS5 induced miR-155. Indeed, miR-155 positively regulates TNF-α to keep it at a high and stable level during the time of initial infection. On the other hand, miR-155 modulates IFNγ production in natural killer cells by Tim-3 to keep the balance between immune clearance and immune injury in CHC patients [82].

### 5.2. MiRNAs Modulating Steatosis

In CHC patients, imbalance in lipid metabolism results in steatosis in liver. Proteins in SREBP family, including SREBP1a, SREBP1c and SREBP2, play important roles in lipid metabolism. Cholesterol homeostasis is specifically regulated by SREBP2. Involvement of cellular miRNAs in modulating lipid homoeostasis and liver disease progression is well characterized. 

#### 5.2.1. miR-27a/b

In response to extracellular signals, the intracellular PI3K/Akt signaling pathway mediates various physiological processes, such as metabolism and cell proliferation, through phosphorylation. HCV infection activates miR-27 (two isoforms, miR27a and 27b), which is a liver-abundant miRNA, through the PI3K pathway [71]. MiR-27 was involved in HCV associated steatosis by two possible mechanisms. First, miR-27a induced lipid accumulation and HDL synthesis by targeting lipid metabolism related transcription factors, retinoid X receptor a (RXRa) and lipid transporter ABCA1. Second, miR-27b was found to target peroxisome proliferator-activated receptor (PPAR-a) and ANGPTL3 leading to lipid droplet (LD) accumulation in HCV infected hepatocytes [71]. In addition, miR-27 significantly suppressed cholesterol synthesis in part by modulating the 3-hydroxy-3-methyl-glutaryl-coenzyme A reductase (HMGCR).

#### 5.2.2. miR-21-5p

De-regulation of PTEN is found in many liver metabolic disorders as PTEN is a master regulator of many metabolic pathways in hepatocytes. HCV-3a core protein enhanced the expression of miR-21-5p, which downregulated PTEN by binding to its 3′UTR. Consequently, increasing of LD size and accumulation of triglycerides and cholesterol esters were found in cells. MiR-21-5p induction by HCV is critical for the HCV-3a core-induced steatosis [83].

#### 5.2.3. miR-c12

Centrosomal protein 350 could sequester PPARα and was the target for miR-c12. Overexpression of miR-c12 in liver cells attenuated triglyceride accumulation and facilitated PPARα mediated transcription of β-oxidation genes [84].

#### 5.2.4. miR-148a and miR-30a

Both miR-148a and miR-30a were downregulated, while their target Tail interacting protein of 47kDa (TIP47) was upregulated in HCV-infected patients [85]. Overexpression of these two microRNAs significantly reduced TIP47 expression and decreased cellular LDs with significant suppression of HCV RNA. Moreover, overexpression of miR-148a and miR-30a also abates the inhibitory effect of HCV on adipocyte differentiation-related proteins [86].

#### 5.2.5. miR-185-5p 

HCV infection or HCV core protein downregulates miR-185-5p expression. MiR-185-5p binds the SREBP2 3′-UTR to inhibit SREBP2 expression, which control cholesterol homeostasis [87].

### 5.3. MiRNAs Promoting Fibrosis

Liver fibrosis results from the excessive accumulation of extracellular matrix proteins, e.g., collagen. Advanced liver fibrosis results in cirrhosis, liver failure, and portal hypertension. Liver fibrosis due to CHC infection is mediated by upregulation of transforming growth factor (TGF)-β. TGF-β is produced by many different cells and is fibrogenic via acting as a chemotactic for fibroblasts, inducing fibroblast proliferation, and enhancing extracellular matrix protein synthesis (Table 4).

#### 5.3.1. miR-19a

Exosomes secreted from HCV-infected hepatocytes with miR-19a were reported to be internalized to hepatic stellate cells (HSCs). Then, miR-19a targeted SOCS3 in HSC and activated the STAT3-mediated TGF-β signaling pathway to enhance fibrosis marker genes [88].

#### 5.3.2. miR-21

HCV activates miR-21 expression [45]. The miR-21 level is positively correlated with fibrotic stage and leads to fibrogenesis via directly targeting SMAD7, a suppressor of TGF-β signaling [89]. MiR-21 can also activate human HSCs during hepatic fibrosis by regulating the PTEN/AKT pathway [83].

#### 5.3.3. miR-200c

MiR-200c, which downregulates fas-associated protein (FAP)-1, was found upregulated in CHC patient. Attenuated FAP-1 leads to the activation of Src kinase signaling, which elevates the expression of collagen and fibroblast growth factor implicated in fibrosis [16].

#### 5.3.4. miR-221/222

The expression of miR-221 and its homolog miR-222 was significantly elevated along the progression of liver fibrosis in CHC patients. The experimental results demonstrated that miR-221/222 bind to the 3′UTR of cyclin-dependent kinase inhibitor 1B (CDKN1B) and inhibit CDKN1B expression. The elevated expression of miR-221/222 is positively correlated with the expression of fibrosis-related α-1 type I collagen and α-smooth muscle actin [90]. Thus, miR-221 plays an important role in liver fibrosis.

#### 5.3.5. miR-16

Increased expression of miR-16 by HCV infection downregulates hepatocyte growth factor (HGF) and Smad7 in the development of liver fibrosis [91].

#### 5.3.6. miR-1273g-3p

MiR-1273g-3p expression was significantly enhanced by HCV infection and correlated with the fibrotic stage induced by HCV. Exogenously expressed miR-1273g-3p could target and reduce PTEN translation, upregulate the expression of α-SMA, Col1A1, and inhibit apoptosis in HSCs [92].

#### 5.3.7. miR-192

Upregulation of miR-192 expression by HCV infection or HCV core protein directly reduced the Zinc Finger E-Box Binding Homeobox 1, which is found to downregulate the expression of TGF-β1. Thus, miR-192 mediates HCV infection-associated fibrogenesis through enhanced TGF-β1 [93].

#### 5.3.8. miR-27

HCV NS3 protein upregulates the expression of miR-27a, which is a pro-fibrotic miRNA. Both miR27a and 27b targeted RXRa and facilitated cell proliferation during the activation of HSCs [94].

#### 5.3.9. miR-10a

HCV infection induced prominent upregulation of miR-10a, which regulates various metabolic genes in liver. In hepatocytes, overexpression of miR-10a inhibited the expression of brain and muscle aryl hydrocarbon receptor nuclear translocator-like 1 (Bmal1) and several lipid synthesis genes (e.g., SREBP1, fatty acid synthase (FASN), and SREBP2). The reduced Bmal1 was associated with the development of liver fibrosis in CHC patients. Therefore, the downregulation of Bmal1 by miR-10a leads to abnormal liver metabolism in cirrhosis [95].

### 5.4. MiRNAs Preventing Fibrosis

#### 5.4.1. miR-29a

Increased miR-29a function inhibits the activation of HSCs and prevents fibrosis [96]. HCV infection suppressed miR-29 expression and miR-29 regulated expression of extracellular matrix proteins. Thus, downregulation of miR-29 by HCV may de-repress the synthesis of extracellular matrix such as collagen during HSC activation [75,97].

#### 5.4.2. miR-449a

Downregulation of miR-449a in patients with HCV infection but not with alcoholic and non-alcoholic liver diseases was identified using gene expression analyses. Upregulation of NOTCH1, the target of miR-449a, was also demonstrated in HCV patients. Furthermore, YKL40 expression, an inflammatory marker increased in patients with chronic liver diseases with fibrosis, is regulated by NOTCH1 in response to TNFα in human hepatocytes. Therefore, miRNA-449a is important for regulating expression of YKL40 by targeting the components of the NOTCH signaling pathway during HCV infection [80]. MiR-449a, along with miR-107 (see miR-107), was found to target components of the interleukin-6 receptor (IL-6R) complex, inhibit IL-6 signaling and modulate CCL2 expression in hepatocytes [98].

#### 5.4.3. miR-107

The expression of CCL2 chemokine was modulated by downregulation of miR-107 and miR-449 (see miR-449) in patients with HCV related liver diseases. Mir-107 was found to target JAK1; therefore, it impaired STAT3 activation in hepatocytes [98].

#### 5.4.4. miR-122

The miR-122 expression is negatively correlated with fibrotic stages. NF-kB-inducing kinase (NIK) is a target of miR-122, whose expression is significantly suppressed by HCV infection [99]. Thus, de-repression of NIK expression by HCV infection dysregulates lipid metabolism. The decreased expression of miR-122 induced by HCV NS3 protein is likely to be a contributing factor to hepatic fibrosis [100].

#### 5.4.5. miR-150

The myeloblastosis transcription factor (C-MYB) is important for the production of smooth-muscle-actin and collagen type I. MiR-150 targeted C-MYB to inhibit the activation of HSCs when overexpressed in cells, which leads to its anti-fibrotic activity [94].

#### 5.4.6. miR-335

miR-335 [101] also has anti-fibrotic activity in cells. MiR-335 suppresses tenascin-C, which is an extracellular matrix glycoprotein associated with cell migration in HSCs [94].

#### 5.4.7. miR-200a

Upregulation of lncRNA-ATB was detected in fibrotic liver tissues. There are common binding sites for miR-200a in lncRNA-ATB. Downregulation of miR-200a and upregulation of β-catenin were found in liver tissues of HCV patients with hepatic fibrosis. Knockdown of lncRNA-ATB could reduce β-catenin expression by increasing the miR-200a and inhibit the activation of LX-2 cells. Thus, in HCV patients, lncRNA-ATB/miR-200a/β-catenin regulatory axis is probably involved in the development of liver fibrosis [102].

### 5.5. MiRNAs Promoting HCC

HCC occurs due to chronic liver inflammation and is associated with chronic viral infection (hepatitis B or C) or exposure to toxins, e.g., aflatoxin. Regulation of miRNAs by HCV infection favors the initiation and progression of HCC. Several reports have demonstrated that miRNAs act as oncogenes or tumor suppressors and exert their functions with induction and suppression of vital biological processes involved in HCC pathogenesis. Identification of these miRNAs help us understand more about their roles in various signaling pathways related to HCC (Table 5).

#### 5.5.1. miR-155

Significant upregulation of miR-155 was detected in HCV patients, which promoted hepatocyte proliferation and tumorigenesis via activation of Wnt signaling pathway [103]. Overexpression of miR-155 has been shown to activate β-catenin and result in an increase in cyclin D1, c-myc, and surviving, which promote cell proliferation. Moreover, miR-155 also reduced a negative Wnt signaling regulator, adenomatous polyposis coli (APC), to enhance hepatocyte proliferation and tumorigenesis [104].

#### 5.5.2. miR-141

DLC-1 is a Rho GTPase-activating protein and also a tumor suppressor. MiR-141 enhanced viral replication and tumorigenesis by inhibiting DLC-1 expression in primary hepatocytes infected with HCV [76].

#### 5.5.3. miR-21

MiR-21 was upregulated after HCV infection [45], and an elevated expression of miR-21 was found in many solid tumors, including HCC. MiR-21 interacts directly with tumor suppressor PTEN, so it facilitates the proliferation, migration, and invasion of hepatoma cells [105].

#### 5.5.4. miR-135a-5p

MiR-135a-5p targeted and reduced hepatic PTPRD expression, a tumor suppressor, in HCV patients. High protein tyrosine phosphatase receptor delta (PTPRD) level is associated with an attenuated transcriptional activity of STAT3. STAT3 is critical for HCV infection. Thus, HCV enhances a STAT3 transcriptional program by inhibiting its regulator PTPRD through increasing miR-135a-5p level. However, a perturbed PTPRD–STAT3 axis might drive malignant progression of HCV-associated liver disease [106].

#### 5.5.5. miR-196a

MiR-196 was reported to reduce the transcriptional activity of SOCS2 factor and then enhance the phosphorylation of JAK2 and STAT5 proteins, which is associated with hepatic lipid metabolism and cancer development. HCV core protein upregulated the miR-196a expression, which enhanced cell proliferation through the induction of the G1-S transition. Furthermore, forkhead box O1 (FOXO1) was directly targeted by miR-196a and was required in the effects of miR-196a on HCC development. Overexpression of FOXO1 significantly reversed the effect of miR-196a on HCC cell proliferation. Thus, elevated miR-196a expression by the HCV core protein can serve as an onco-microRNA in HCV-induced cell proliferation through reducing the FOXO1 expression [107].

#### 5.5.6. Other miRNAs

The topological analysis of miRNA-Hub gene network has identified the key hub miRNAs. The most important hub miRNAs, which are positively correlated with chronic HCV and HCC samples, include miR-34a, miR-155, miR-24, miR-744 and miR-92a [113].

### 5.6. MiRNAs Preventing HCC

#### 5.6.1. miR-138

The mature HCV core protein reduces miR-138 expression. Indeed, miR-138 was often found to be downregulated in HCC. In cells transfected with miR-138, cell proliferation was suppressed due to reduced TERT activity, which led to cell senescence. Thus, in the presence of HCV core protein, a reduced level of miR-138 may reactivate TERT expression and possibly lead to cancer development in HCV-infected cells [108].

#### 5.6.2. miR-203

MiR-203 silencing by hypermethylation was found in primary HCC patients compared with healthy controls. HCV core protein has been demonstrated to significantly suppress the expression of miR-203a, which targets SNAL2 and induces EMT to enhance HCC aggressiveness. The oncogene ADAM Metallopeptidase Domain 9 (ADAM9) and lncRNA HULC are also modulated by miR-203 to block the invasion and migration of cancer cells via targeting ATP Binding Cassette Subfamily E Member 1 (ABCE1) factor. Moreover, miR-203 inhibits cell proliferation by reducing the anti-apoptotic protein survivin [109].

#### 5.6.3. miR-30c

MiR-30c expression was markedly lower in HCC patients with HCV than in HCC patients without HCV. HCV core protein has been demonstrated to significantly decrease the expression of miR-30c, which targets SNAL1 and induce EMT. MiR-30c acts as a tumor suppressor, and it inhibits TGF-β-induced Serpine 1 and/or B-Cell CLL/Lymphoma 9 Protein (BCL9) to suppresses cell growth [109].

#### 5.6.4. miR-122

It is well established that miR-122 is involved in HCV infection and inhibition of HCC [21]. MiR-122 suppresses HCC by targeting genes implicated in cancer development, such as ADAM Metallopeptidase Domain 17. MiR-122 also targets Cyclin G1, a negative regulator of P53, to enhance P53 expression. Besides, miR-122 reduces the level of insulin growth factor 1 receptor (IGF-1R). Reduction of IGF-1R/AKT signaling maintains glycogen synthase kinase-3 beta activity and results in the suppression of cyclin D1 expression and cell proliferation. MiR-122 also directly targets WNT1, so it functions as an inhibitor in the WNT/β-catenin signaling pathway.

#### 5.6.5. miR-152

WNT/β-catenin signaling pathway is very conserved and involved in cell proliferation. HCV core protein inhibited the expression miR-152, which directly targets the 3′-UTR of the WNT1, an activating ligand for β-catenin pathway. Then, de-repressing Wnt1-mediated signaling promoted G1-S transition, cell proliferation, and colony formation. Therefore, miR-152 is thought to be a tumor suppressor. Multiple HCC cell lines transfected with miR-152 induced apoptosis and showed decreased cell proliferation and mobility due to inactivation of ERK and AKT pathways. MiR-152 also directly targets rhotekin, a tumor-promoting protein, to reduce tumor development [110].

#### 5.6.6. miR-491

Chronic HCV infection reduced the expression of miR-491. miR-491 inhibits PI3K-Akt pro-survival pathway and its reduced expression promotes tumorigenesis [79].

#### 5.6.7. miR-181c

Suppression of miR-181c expression was detected in HCV infected hepatocytes. The ATM protein is a target of miR-181c. HCV infection downregulated miR-181c expression in hepatocytes and resulted in ATM upregulation, which inhibits apoptosis for the promotion of cell cycle progression [65].

#### 5.6.8. miR-124

MiR-124 expression is often downregulated in HCC. HCV core protein also suppressed miR-124 expression. MiR-124 exerts its effects through regulation of SMYD3, a regulator of the oncogenes c-Myc and MMP9 [111].

#### 5.6.9. miR-148a-3p

The role of miR-148a-3p, which is downregulated in HCV-infected HCC cells, has been investigated in HCV-related cancers. MiR-148a-3p targets c-Jun mRNA to suppress the protein expression, and then reduced MAPK signaling pathway [112].

#### 5.6.10. miR-503

MiR-503 has an anti-tumor effect. HCV NS5A protein suppresses the level of miR-503, which may contribute to carcinogenesis in hepatocytes. MiR-503 has been shown to target protein arginine methyltransferase 1 and WEE1 G2 Checkpoint Kinase to prevent EMT in HepG2 cells [64].

### 5.7. MiRNAs in Other HCV-Related Diseases

#### 5.7.1. The Roles of miRNAs in HCV-Related Diabetes

The roles of miRNAs in HCV-related diabetes are poorly characterized. Increasing evidence suggested that miRNAs in circulation can modulate gene expression in remote tissues, which may be involved in the development of diabetes [114,115]. A previous study demonstrated that circulating miR-122 level is associated with the risk of developing metabolic syndrome and type II diabetes in the general population [116].

In contrast, circulating miR-155 levels were markedly reduced in diabetic HCV patients compared to those in nondiabetic HCV patients. Interestingly, the circulating levels of miR-155 were negatively correlated with homeostatic model assessment for insulin resistance, fasting blood glucose and HbA1c levels. This suggests an involvement of miR-155 in the pathogenesis of insulin resistance caused by HCV infection [117].

#### 5.7.2. The Roles of miRNAs in HCV-Related Cryoglobulinemic Vasculitis

HCV is also a lymphotropic virus that leads to lymphoproliferative disorders including cryoglobulinemic vasculitis (CV). The molecular mechanisms regarding how HCV induces CV remain elusive. All members of the miR17-92 cluster were markedly increased in CV patients compared to those without CV and significantly decreased in those who achieved remission from vasculitis after HCV eradication. This may suggest an involvement of miR17-92 in the pathogenesis of HCV-related CV [118].

## 6. MiRNA as Potential Biomarkers for HCV-Related Diseases

The expression of intracellular miRNAs is dysregulated after HCV infection. These altered miRNAs have been found to modulate HCV propagation and involve in HCV-related diseases. These miRNAs may serve as alternative biomarkers for HCV infection and/or HCV-related diseases. Using miRNAs as a diagnostic biomarker for HCV infection may not be practical because the commercially available diagnostic kits for HCV infection already have high specificities and sensitivities [119]. However, available diagnostic markers (e.g., α-fetoprotein) show poor performance in early diagnosis of HCC. This has encouraged researchers to search for novel potential biomarkers of HCC. Early diagnosis of HCC can significantly improve the overall survival of patients. To be disease-related markers, miRNAs have the advantage over proteins as they can be amplified easily, e.g., by qRT-PCR. A number of dysregulated miRNAs in HCV-related diseases have been suggested to be potential biomarkers. miRNAs that are secreted into blood, e.g., as exosomes, can be easily detected. Therefore, those secreted miRNAs could be potential markers for HCV-related diseases, especially for HCC. Several serum/plasma miRNAs such as miR-21, miR-122, mi-125a/b, miR-199a/b, miR-221, miR-222, miR-223, miR-224 summarized by a previous review article might serve as biomarkers for early diagnosis/prognosis of HBV- or HCV-related HCC. However, neither definitive results nor well-defined panels of miRNAs were obtained [120].

Recently, more circulatory miRNAs have been reported to be potential biomarkers for HCV-related diseases. For example, (1) fibrotic HCV patients have increased miRNA-221 [121], miRNA-222 [121], miR-21 [122], miR-122 [122] and miR-199 [123] and decreased miR-484 [124] and miR-448 [123], compared to non-fibrotic HCV patients; (2) Cirrhotic HCV patients have increased miR-524-5p [124], miR-615-5p [124], and miR-542 [125] and decreased miR-182 [126] and miR-150 [126], compared to noncirrhotic HCV patients; (3) HCV-related HCC patients have increased miR-486-5p [127], miR-224 [128,129], miR-484 [124], miR-155 [104,130], miR-183 [104], miR-665 [130], miR-331-3p [129,131], miR-494-3p [129], miR-301 [132], miR-1269 [133], miR-483-5p [134], miR-133a [134] and miR-542 [125] and decreased miR-182 [126], miR-150 [126], miR 9-3p [135], miR-185-5p [129], miR-23b-3p [129,131], let-7 family members [136,137], miR-139 [138], miR-145 [137,138], miR-199a [138], miR-143 [137] and miR-152 [139], compared to non-HCC HCV patients. The reported expression levels of miRNA-122 [128,129,140], miR-221 [125,140,141], in HCV-related HCC patients from different studies are contradictory. Further studies with more samples are needed.

When more information is available, using several miRNAs together to detect HCV-related diseases will probably increase the sensitivity and specificity of this biomarker as the diagnostic tool.

## 7. MiRNAs as Potential Targets for Anti-HCV Therapies

In addition to targeting viral proteins (i.e., DAAs), anti-HCV therapies could also aim at the cellular factors modulating HCV replication. Because of its ability to enhance HCV replication profoundly and because its seed sequences are highly conserved across all HCV genotypes, miR-122 has become a potential therapeutic target for chronic HCV infections [21,142]. Miravirsenis, a locked nucleic-acid-modified DNA phosphorothioate antisense oligonucleotide, is designed to interact with and then block the function of miR-122. In phase I and II clinical trials, miravirsen induced a sustained reduction dose-dependently in HCV viral loads [142]. Moreover, when combined with other DAAs, miravirsen has an additive effect [143]. However, all subjects had a virologic rebound after treatment [144]. Further studies suggest that the occurrence of HCV variants can grow in nonhepatic cells in a miR-122-independent manner. These HCV variants may induce resistance to miR-122-antagonist therapy, i.e., miravirsen [145,146].

Due to the success of various DAAs, it may not be practical to use miRNAs as the targets for anti-HCV therapies. On the other hand, miRNAs are potential therapeutic targets for HCC treatment [4].

## 8. Conclusions

Cellular miRNA expression is dysregulated during HCV infection. Many of these miRNAs have been characterized. Indeed, these miRNAs play significant roles in HCV infection and disease pathophysiology. Some of these miRNAs secreted as exosomes could be potential minimally invasive biomarkers for HCV-related diseases, especially HCC. This is important because the chance of HCC occurrence and recurrence remains even after HCV is eradicated by DAA. In addition to DAAs, host-targeting antivirals have been studied in the HCV therapy, e.g., therapeutic depletion of miR-122 for anti-HCV treatment. Some of the miRNAs involved in HCV-related diseases might be therapeutic targets. Thus, gene therapies such as miRNA antagonists might become a common treatment for HCV-related diseases such as HCC in the future.

Hundreds of cellular proteins implicated in the HCV life cycle have been characterized in the past thirty years of study [3]. The miRNAs that regulate the expression of these cellular proteins involved in HCV lifecycle should also modulate HCV replication, and/or even HCV-related disease progression. Therefore, there should be plenty of miRNAs, identified and unidentified, involved in HCV replication and pathogenesis. These miRNAs are potential targets for future research and development.

## Figures and Tables

**Figure 1 viruses-14-01776-f001:**
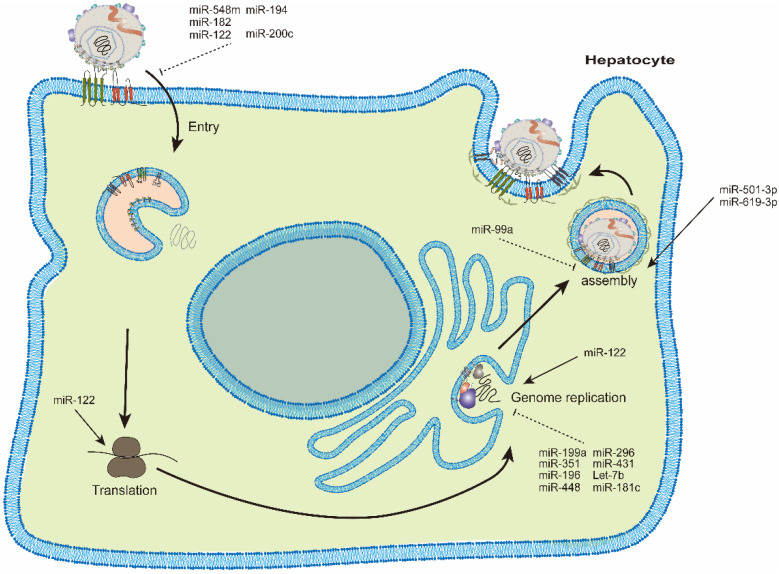
Various microRNAs modulate hepatitis C virus replication in different stages of viral lifecycle. Several miRNAs suppress HCV entry, such as miR-548m, miR-194, miR-182, miR-122 and miR-200c. Several miRNAs such as miR-196, miR-296, miR-351, miR-431, miR-448, miR-199a, let-7b and miR-181c suppress while miR-122 facilitates HCV genome replication. MiR-99a suppresses while miR-501-3p and miR-619-3p facilitate HCV assembly.

**Figure 2 viruses-14-01776-f002:**
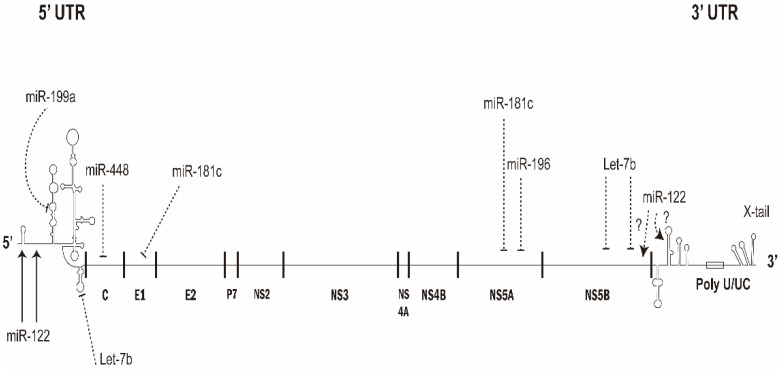
Target sites of various microRNAs on HCV genomic RNA. The 5′UTR of HCV genomic RNA contains the target sites for miR-122, miR-199a* and let-7b; core gene for miR-448; E1 for miR-181c; NS5A for miR-181c and miR-196; and NS5B for let-7b. After binding to the target sites of HCV genomic RNA, several miRNAs such as miR-196, miR-448, miR-199a, let-7b and miR-181c suppress while miR-122 facilitates HCV genome replication and/or translation.

**Table 1 viruses-14-01776-t001:** miRNAs involved in different stages of HCV life cycle.

miRNA Name	Regulatory Mechanism	Biologic Effects
miR-548m [12]	Targets and suppresses CD81 expression	Hinders HCV entry
miR-194 [13]	Targets and suppresses CD81 expression	Hinders HCV entry
miR-182 [14]	Targets and suppresses CLDN1 expression	Hinders HCV entry
miR-122 [15]	Targets and suppresses OCLN expression	Hinders HCV entry
miR-200c [16]	Targets and suppresses OCLN expression	Hinders HCV entry
miR-196, miR-296, miR-351, miR-431 and miR-448 [17]	Interacts with HCV genome directly	Inhibits HCV replication
miR-199a [18]	Interacts with 5’-UTR of HCV genome directly	Inhibits HCV replication
let-7b [19]	Interacts with the NS5B coding region and 5’-UTR of HCV genome directly	Inhibits HCV replication
miR-181c [20]	Interacts with the E1 and NS5A coding regions of HCV genome directly	Inhibits HCV replication
miR-122 [21]	Interacts with 5’-UTR of HCV genome directly	Facilitates HCV translation and/or genome replication
miR-99a [22]	Reduces intracellular lipid accumulation	Suppresses HCV replication and packaging
miR-501-3p and miR-619-3p [23]	Targets and suppresses OGT expression	Facilitates HCV assembly

**Table 2 viruses-14-01776-t002:** miRNAs involved in the interferon pathways.

miRNA Name	Regulatory Mechanism	Biologic Effects
miR16 [44]	Downregulates SMAD7	Inhibits IFN production
miR-21 [45]	Inhibits NF-kB, MyD88 and IRAK1	Suppresses IFN production and signaling
miR-208b and miR-499a-5p [46]	Suppresses IFNL2 and IFNL3; Reduces IFNAR1 expression	Suppresses type III IFN; Attenuates type I IFN signaling
miR-93-5p [47]	Targets IFNAR1	Inhibits the IFN signaling
miR-373 [48,49]	Targets JAK1, IRF9 and IRF5	Reduces the IFN signaling
miR-135a [50]	Targets RIPK2, MYD88, and CXCL12	Suppresses IFN signaling
miR-758 [51]	Suppresses TLR3 and TLR7	Reduces IFN production and signaling
miR- 125a [52]	Targets MAVS and TRAF6	Reduces the IFN signaling
miR-942 [53]	Targets ISG12a	Reduces the IFN effect
miR-221 [54]	Inhibits the expression of SOCS1 and SOCS3	Increases IFN-α activity
miR-30 [55]	Targets SOCS1 and SOCS3	Enhances cytokine signaling
Let-7b [56]	Targets SOCS1, ATG12 and IKKα	Enhances IFN expression
miR-122 [57]	Targets MERTK, FGFR1 and IGF1R	Enhances IFN signaling
miR-29c [58]	Targets and suppresses STAT3 expression	Enhances type I IFN response

**Table 3 viruses-14-01776-t003:** miRNAs involved in HCV replication.

miRNA Name	Regulatory Mechanism	Biologic Effects
miR-125b-5p [62]	Targets HuR	Reduces HCV replication
miR-196a [60]	Reduces Bach1 expression	Inhibits HCV replication
Let-7c [63]	Targets Bach1	Reduce HCV replication
miR-503 [64]	Targets and reduces BCL-2 expression	Enhances apoptosis
miR-181c [65]	Targets and reduces ATM expression	Promotes apoptosis of HCV infected hepatocytes
miR-130a [66,67,68,69]		Contradictory results were reported
miR-27a [70,71,72,73]	Reduces the expression of many genes involved in lipid metabolism including ApoA1, ApoB100 and ApoE3	Reduces the production of infectious HCV particles
miR-185 [74]	Targets GALNT	Suppresses HCV replication
miR-29 [75]	Not known	Reduces HCV replication
miR-141 [76]	Targets and reduces DLC-1 expression	Increases HCV replication
miR-320c, miR-483 [77]	The PI3K/Akt, MAPK and NF-κB signaling pathway were targeted by miR-320c and miR-483-5p	Play roles in immune evasion and cell survival
miR-122 [78]	Downregulation of HO-1	Facilitate HCV replication
miR-491 [79]	Targets PI3K/Akt pathway	Help HCV entry

**Table 4 viruses-14-01776-t004:** miRNAs modulating HCV-related fibrosis.

miRNA Name	Regulatory Mechanism	Biologic Effects
miR-19a [88]	Targets SOCS3 in HSC and activates the TGF-β signaling	Promotes fibrosis
miR-21 [83,89]	Targets SMAD7 and activates the TGF-β signaling Activates HSCs via the PTEN/AKT pathway	Promotes fibrosis
miR-200c [16]	Enhances the expression of collagen and fibroblast growth factor	Promotes fibrosis
miR-221/222 [90]	Inhibits expression of CDKN1B	Promotes fibrosis
miR-16 [91]	Downregulates HGF and Smad7	Promotes fibrosis
miR-1273g-3p [92]	Inhibits PTEN, increases the expression of a-SMA, Col1A1, and reduces apoptosis in HSCs	Promotes fibrosis
miR-192 [93]	Represses ZEB1 and thus enhances TGF-β1	Promotes fibrosis
miR-27 [94]	Promotes cell proliferation during HSCs activation	Promotes fibrosis
miR-10a [95]	Downregulates Bmal1 expression	Promotes fibrosis
miR-29a [75,96,97]	Suppresses the activation of HSCs	Reduces expression of extracellular matrix proteins Prevents fibrosis
miR-449a [80,98]	Downregulates NOTCH1 Targets IL-6R and impairs STAT3 activation	Prevents fibrosis
miR-107 [98]	Targets JAK1 and impairs STAT3 activation	Prevents fibrosis
miR-122 [99,100]	Targets NIK	Prevents fibrosis
miR-150 [94]	Inhibits the activation of HSCs by targeting the C-MYB	Prevents fibrosis
miR-335 [94,101]	Inhibits tenascin-C involved in cell migration	Prevents fibrosis
miR-200a [102]	Inhibitsβ-catenin expression	Prevents fibrosis

**Table 5 viruses-14-01776-t005:** miRNAs modulating HCV-related hepatocellular carcinoma.

miRNA Name	Regulatory Mechanism	Biologic Effects
miR-155 [103,104]	Activates Wnt signaling	Promotes hepatocyte proliferation and tumorigenesis
miR-141 [76]	Reduces DLC-1 expression	Promotes tumorigenesis
miR-21 [105]	Interacts with PTEN	Enhances cell proliferation, migration, and invasion of hepatoma cells
miR-135a-5p [106]	Inhibits PTPRD expression	Promotes tumorigenesis
miR-196a [107]	Inhibits FOXO1 expression	Enhances cell proliferation
miR-138 [108]	Decreases TERT activity	Suppresses cell proliferation, and induces cell senescence
miR-203 [109]	Targets SNAL2; Modulates ADAM9; Suppresses Survivin	Suppresses the invasion and migration of cancer cells
miR-30c [109]	Targets SNAL1; Inhibits Serpine 1 and BCL9	Suppresses cell growth
miR-122 [21]	Targets cyclin G1, ADAM Metallopeptidase Domain 17 and WNT1	Prevents tumorigenesis
miR-152 [110]	Targets WNT1	Suppresses cell proliferation and motility
miR-491 [79]	Inhibits PI3K-Akt	Prevents tumorigenesis
miR-181c [65]	Targets ATM	Prevents cell cycle progression
miR-124 [111]	Modulates SMYD3	Prevents tumorigenesis
miR-148a-3p [112]	Targets c-Jun	Prevents tumorigenesis
miR-503 [64]	Targets protein arginine methyltransferase 1 and WEE1 G2 Checkpoint Kinase	Blocks EMT

## Data Availability

Not applicable.

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
