# Peer review of "Roles of microRNAs in Hepatitis C Virus Replication and Pathogenesis"

_viruses, 2022, doi:10.3390/v14081776_

Round 1

Reviewer 1 Report

This review provides a comprehensive overview of the variety of micro-RNAs somehow linked to HCV replication or pathogenesis in literature. While the comprehensiveness in this aspect is an advantage, it severely affects readability and focus. Most miRs are addressed in subchapters of two sentences and therefore very superficial. A fluent text with associated tables as a summary (tables as in the “fibrosis/HCC” section) would be preferable. While the lack of detail is acceptable for 99% of miRs described, it is not for miR-122 which is exceptional in all regards. This aspect clearly requires a distinct far more extensive subchapter, not only referencing a review, but also taking into account far more primary literature, starting from its discovery to more recent novel findings regarding its mode of action and pathogenesis aspects. Since miR-122 is the first thought every HCV scientist will have, it clearly needs far more attention. While it is not possible to include all ca. 300 papers on the topic, a better summary of the known mechanisms as well as a comprehensive citation of novel literature is in my view essential. In turn, other aspects (e.g. Interferon modulated miRs) could be condensed substantially, e.g. by listing them in a table but in turn reducing redundant text, but rather summarizing the aspect (….are involved in modulating … by various mechanisms). At the current stage, this review does not provide any guidance towards important and less important aspects, this needs improvement. For example, factors that have been found upon overexpression of individual HCV proteins only (e.g.  ref. 47) need to be regarded more critically as observations in more physiological models and in vivo.

Specific comments:

The miR-122 chapter needs far more attention and primary literature. Discovery: PMID: 16141076; phenotype of knockout-mice PMID: 22820290, sponging/pathogenesis (PMID: 25768906) recent mechanistic studies starting 2018, including a mechanistic understanding of its roles in translation and of DUSP11 in genome stability: PMID: 34908444; PMID: 34824224; PMID: 34385308; PMID: 32810257; PMID: 32574204; PMID: 30941417; PMID: 30053137; PMID: 30038017; PMID: 29973597; PMID: 29672716; PMID: 29669014
Comparison to Hepaciviruses/convervation: e.g. PMID: 33713356

List all the different miRs found in literature in tables, as exemplified in tables 1 and 2. In turn, reduce the summarizing text.

Readability needs improvement by reducing the fragmentation of the text.

Line 198: …is reported: please add citation

Author Response

  Specific comments:

The miR-122 chapter needs far more attention and primary literature. Discovery: PMID: 16141076; phenotype of knockout-mice PMID: 22820290, sponging/pathogenesis (PMID: 25768906) recent mechanistic studies starting 2018, including a mechanistic understanding of its roles in translation and of DUSP11 in genome stability: PMID: 34908444; PMID: 34824224; PMID: 34385308; PMID: 32810257; PMID: 32574204; PMID: 30941417; PMID: 30053137; PMID: 30038017; PMID: 29973597; PMID: 29672716; PMID: 29669014
Comparison to Hepaciviruses/convervation: e.g. PMID: 33713356

Responses: Thanks for the suggestions! All of these suggested references are included in the revised manuscript [pages 4-5, marked in red].

List all the different miRs found in literature in tables, as exemplified in tables 1 and 2. In turn, reduce the summarizing text.

Responses: Thanks for the suggestions! Tables 1 and 2 are added as suggested. Original tables 1-2 are changed to 3-4.

Readability needs improvement by reducing the fragmentation of the text.

Responses: Thanks for the suggestions! We have also checked the manuscript.

Line 198: …is reported: please add citation

Responses: Thanks for the suggestions! One reference is added as suggested [marked in red].

Reviewer 2 Report

This review by Hui-Chun-Li and colleagues provides a good overview about the various functions of miRNAs in HCV life cycle. The review is well structured and clearly written. It is a helpful concept to describe various aspects of HCV life cycle and HCV-associated pathogenesis and to report  the function of various miRNAs in this context. The relevant and actual literature is cited. Conflicting data are in  a balanced way described.

There are two minor points:

Line 12:” HCV is considered as an intracellular pathogen…. Or line 88…”HCV releies heavily on its host cell to infect….” This described general characteristics of viruses—it should be  modified

Line 60:  the HCV release pathway is a little bit more complex as described here. It`s not the focus of the review, but this  should be modified

The authors could include an additional paragraph about miRNA as targets for novel antiviral  thererapies.

Author Response

Line 12:” HCV is considered as an intracellular pathogen…. Or line 88…”HCV releies heavily on its host cell to infect….” This described general characteristics of viruses—it should be  modified

Responses: Thanks for the suggestions! Sentences are revised as suggested [marked in red].

Line 60:  the HCV release pathway is a little bit more complex as described here. It`s not the focus of the review, but this  should be modified

Responses: Thanks for the suggestions! Sentences are modified as suggested [marked in red].

 The authors could include an additional paragraph about miRNA as targets for novel antiviral  thererapies.

Responses: Thanks for the suggestions! Additional paragraph about miRNA as targets was added as suggested [pages 18-19, marked in red].

Reviewer 3 Report

In this current review report authors have outlined clearly ‘Roles of microRNAs in hepatitis C virus replication and pathogenesis’.  The role of differentially expressed miRNAs at stages of HCV pathogenesis and extra hepatic diseases are summarized comprehensively. Although this review report is important for the target audience to highlight the importance of miRNAs in HCV pathogenesis, the significance of HCV genotypes pathogenesis and therapeutic outcome is important. This report has limited or no mention of miRNAs role in viral genotypes. I have a minor comment to be clarified before publication.

Why authors omitted HCV genotypes pathogenesis in this review. Also, reported miRNAs in different stages of viral infection are HCV genotype agnostic? you may amend revisions in your revised manuscript.

Author Response

In this current review report authors have outlined clearly ‘Roles of microRNAs in hepatitis C virus replication and pathogenesis’.  The role of differentially expressed miRNAs at stages of HCV pathogenesis and extra hepatic diseases are summarized comprehensively. Although this review report is important for the target audience to highlight the importance of miRNAs in HCV pathogenesis, the significance of HCV genotypes pathogenesis and therapeutic outcome is important. This report has limited or no mention of miRNAs role in viral genotypes. I have a minor comment to be clarified before publication.

Why authors omitted HCV genotypes pathogenesis in this review. Also, reported miRNAs in different stages of viral infection are HCV genotype agnostic? you may amend revisions in your revised manuscript.

Responses: Thanks for the suggestions. An additional paragraph regarding HCV genotypes was added in the revised manuscript [Page 3 marked in red].

Round 2

Reviewer 1 Report

The authors addressed most of my comments and the manuscript is clearly improved, particularly the additional tables are helpful. Therefore I would suggest to add an additional table on the miRs involved in the HCV replication cycle (p3-5). This will overall render this review a comprehensive resource.

Author Response

The authors addressed most of my comments and the manuscript is clearly improved, particularly the additional tables are helpful. Therefore I would suggest to add an additional table on the miRs involved in the HCV replication cycle (p3-5). This will overall render this review a comprehensive resource.

Responses: Thanks for the suggestions! Tables 1 is added as suggested (pages 3-4). Original tables 1-4 are changed to 2-5.